# TEMPORAL DIFFERENCE
# VARIATIONAL AUTO-ENCODER

**Karol Gregor, George Papamakarios, Frederic Besse, Lars Buesing, Théophane Weber**
DeepMind
`{karolg, gpapamak, fbesse, lbuesing, theophane}@google.com`

## ABSTRACT

To act and plan in complex environments, we posit that agents should have a mental simulator of the world with three characteristics: (a) it should build an abstract state representing the condition of the world; (b) it should form a belief which represents uncertainty on the world; (c) it should go beyond simple step-by-step simulation, and exhibit temporal abstraction. Motivated by the absence of a model satisfying all these requirements, we propose TD-VAE, a generative sequence model that learns representations containing explicit beliefs about states several steps into the future, and that can be rolled out directly without single-step transitions. TD-VAE is trained on pairs of temporally separated time points, using an analogue of temporal difference learning used in reinforcement learning.

## 1 INTRODUCTION

Generative models of sequential data have received a lot of attention, due to their wide applicability in domains such as speech synthesis (van den Oord et al., 2016a; 2017), neural translation (Bahdanau et al., 2014), image captioning (Xu et al., 2015), and many others. Different application domains will often have different requirements (e.g. long term coherence, sample quality, abstraction learning, etc.), which in turn will drive the choice of the architecture and training algorithm.

Of particular interest to this paper is the problem of reinforcement learning in partially observed environments, where, in order to act and explore optimally, agents need to build a representation of the uncertainty about the world, computed from the information they have gathered so far. While an agent endowed with memory could in principle learn such a representation implicitly through model-free reinforcement learning, in many situations the reinforcement signal may be too weak to quickly learn such a representation in a way which would generalize to a collection of tasks.

Furthermore, in order to plan in a model-based fashion, an agent needs to be able to imagine distant futures which are consistent with the agent's past. In many situations however, planning step-by-step is not a cognitively or computationally realistic approach.

To successfully address an application such as the above, we argue that a model of the agent's experience should exhibit the following properties:

- The model should learn an abstract *state representation* of the data and be capable of making predictions at the state level, not just the observation level.

- The model should learn a *belief state*, i.e. a deterministic, coded representation of the filtering posterior of the state given all the observations up to a given time. A belief state contains all the information an agent has about the state of the world and thus about how to act optimally.

- The model should exhibit *temporal abstraction*, both by making 'jumpy' predictions (predictions several time steps into the future), and by being able to learn from temporally separated time points without backpropagating through the entire time interval.

To our knowledge, no model in the literature meets these requirements. In this paper, we develop a new model and associated training algorithm, called *Temporal Difference Variational Auto-Encoder* (TD-VAE), which meets all of the above requirements. We first develop TD-VAE in the sequential, non-jumpy case, by using a modified evidence lower bound (ELBO) for stochastic state space models

(Krishnan et al., 2015; Fraccaro et al., 2016; Buesing et al., 2018) which relies on jointly training a filtering posterior and a local smoothing posterior. We demonstrate that on a simple task, this new inference network and associated lower bound lead to improved likelihood compared to methods classically used to train deep state-space models.

Following the intuition given by the sequential TD-VAE, we develop the full TD-VAE model, which learns from temporally extended data by making jumpy predictions into the future. We show it can be used to train consistent jumpy simulators of complex 3D environments. Finally, we illustrate how training a filtering a posterior leads to the computation of a neural belief state with good representation of the uncertainty on the state of the environment.

## 2 MODEL DESIDERATA

### 2.1 CONSTRUCTION OF A LATENT STATE-SPACE

**Autoregressive models**. One of the simplest way to model sequential data $(x_1, \ldots, x_T)$ is to use the chain rule to decompose the joint sequence likelihood as a product of conditional probabilities, i.e. $\log p(x_1, \ldots, x_T) = \sum_t \log p(x_t \mid x_1, \ldots, x_{t-1})$. This formula can be used to train an autoregressive model of data, by combining an RNN which aggregates information from the past (recursively computing an internal state $h_t = f(h_{t-1}, x_t)$) with a conditional generative model which can score the data $x_t$ given the context $h_t$. This idea is used in handwriting synthesis (Graves, 2013), density estimation (Uria et al., 2016), image synthesis (van den Oord et al., 2016b), audio synthesis (van den Oord et al., 2017), video synthesis (Kalchbrenner et al., 2016), generative recall tasks (Gemici et al., 2017), and environment modeling (Oh et al., 2015; Chiappa et al., 2017).

While these models are conceptually simple and easy to train, one potential weakness is that they only make predictions in the original observation space, and don't learn a compressed representation of data. As a result, these models tend to be computationally heavy (for video prediction, they constantly decode and re-encode single video frames). Furthermore, the model can be computationally unstable at test time since it is trained as a next step model (the RNN encoding real data), but at test time it feeds back its prediction into the RNN. Various methods have been used to alleviate this issue (Bengio et al., 2015; Lamb et al., 2016; Goyal et al., 2017; Amos et al., 2018).

**State-space models**. An alternative to autoregressive models are models which operate on a higher level of abstraction, and use latent variables to model stochastic transitions between *states* (grounded by observation-level predictions). This enables to sample state-to-state transitions only, without needing to render the observations, which can be faster and more conceptually appealing. They generally consist of decoder or prior networks, which detail the generative process of states and observations, and encoder or posterior networks, which estimate the distribution of latents given the observed data. There is a large amount of recent work on these type of models, which differ in the precise wiring of model components (Bayer & Osendorfer, 2014; Chung et al., 2015; Krishnan et al., 2015; Archer et al., 2015; Fraccaro et al., 2016; Liu et al., 2017; Serban et al., 2017; Buesing et al., 2018; Lee et al., 2018; Ha & Schmidhuber, 2018).

Let $\mathbf{z} = (z_1, \ldots, z_T)$ be a state sequence and $\mathbf{x} = (x_1, \ldots, x_T)$ an observation sequence. We assume a general form of state-space model, where the joint state and observation likelihood can be written as $p(\mathbf{x}, \mathbf{z}) = \prod_t p(z_t \mid z_{t-1}) p(x_t \mid z_t)$.[1] These models are commonly trained with a VAE-inspired bound, by computing a posterior $q(\mathbf{z} \mid \mathbf{x})$ over the states given the observations. Often, the posterior is decomposed autoregressively: $q(\mathbf{z} \mid \mathbf{x}) = \prod_t q(z_t \mid z_{t-1}, \phi_t(\mathbf{x}))$, where $\phi_t$ is a function of $(x_1, \ldots, x_t)$ for filtering posteriors or the entire sequence $\mathbf{x}$ for smoothing posteriors. This leads to the following lower bound:

$$\log p(\mathbf{x}) \geq \mathbb{E}_{\mathbf{z} \sim q(\mathbf{z} \mid \mathbf{x})} \left[ \sum_t \log p(x_t \mid z_t) + \log p(z_t \mid z_{t-1}) - \log q(z_t \mid z_{t-1}, \phi_t(\mathbf{x})) \right]. \quad (1)$$

---

[1]For notational simplicity, $p(z_1 \mid z_0) = p(z_1)$. Also note the conditional distributions could be very complex, using additional latent variables, flow models, or implicit models (for instance, if a deterministic RNN with stochastic inputs is used in the decoder).

## 2.2 ONLINE CREATION OF BELIEF STATE.

A key feature of sequential models of data is that they allow to reason about the conditional distribution of the future given the past: $p(x_{t+1}, \ldots, x_T \mid x_1, \ldots, x_t)$. For reinforcement learning in partially observed environments, this distribution governs the distribution of returns given past observations, and as such, it is sufficient to derive the optimal policy. For generative sequence modeling, it enables conditional generation of data given a context sequence. For this reason, it is desirable to compute sufficient statistics $b_t = b_t(x_1, \ldots, x_t)$ of the future given the past, which allow to rewrite the conditional distribution as $p(x_{t+1}, \ldots, x_T \mid x_1, \ldots, x_t) \approx p(x_{t+1}, \ldots, x_T \mid b_t)$. For an autoregressive model as described in section 2.1, the internal RNN state $h_t$ can immediately be identified as the desired sufficient statistics $b_t$. However, for the reasons mentioned in the previous section, we would like to identify an equivalent quantity for a state-space model.

For a state-space model, the filtering distribution $p(z_t \mid x_1, \ldots, x_t)$, also known as the belief state in reinforcement learning, is sufficient to compute the conditional future distribution, due to the Markov assumption underlying the state-space model and the following derivation:

$$p(x_{t+1}, \ldots, x_T \mid x_1, \ldots, x_t) = \int p(z_t \mid x_1, \ldots, x_t) p(x_{t+1}, \ldots, x_T \mid z_t) \, \mathrm{d}z_t. \tag{2}$$

Thus, if we train a network that extracts a code $b_t$ from $(x_1, \ldots, x_t)$ so that $p(z_t \mid x_1, \ldots, x_t) \approx p(z_t \mid b_t)$, $b_t$ would contain all the information about the state of the world the agent has, and would effectively form a neural belief state, i.e. a code fully characterizing the filtering distribution.

Classical training of state-space model does not compute a belief state: by computing a joint, autoregressive posterior $q(\mathbf{z} \mid \mathbf{x}) = \prod_t q(z_t \mid z_{t-1}, \mathbf{x})$, some of the uncertainty about the marginal posterior of $z_t$ may be 'leaked' in the sample $z_{t-1}$. Since that sample is stochastic, to obtain all information from $(x_1, \ldots, x_t)$ about $z_t$, we would need to re-sample $z_{t-1}$, which would in turn require re-sampling $z_{t-2}$ all the way to $z_1$.

While the notion of a belief state itself and its connection to optimal policies in POMDPs is well known (Astrom, 1965; Kaelbling et al., 1998; Hauskrecht, 2000), it has often been restricted to the tabular case (Markov chain), and little work investigates computing belief states for learned deep models. A notable exception is (Igl et al., 2018), which uses a neural form of particle filtering, and represents the belief state more explicitly as a weighted collection of particles. Related to our definition of belief states as sufficient statistics is the notion of predictive state representations (PSRs) (Littman & Sutton, 2002); see also (Venkatraman et al., 2017) for a model that learns PSRs which, combined with a decoder, can predict future observations.

Our last requirement for the model is that of temporal abstraction. We postpone the discussion of this aspect until section 4.

## 3 BELIEF-STATE-BASED ELBO FOR SEQUENTIAL TD-VAE

In this section, we develop a sequential model that satisfies the requirements given in the previous section, namely (a) it constructs a *latent state-space*, and (b) it creates a online *belief state*. We consider an arbitrary state space model with joint latent and observable likelihood given by $p(\mathbf{x}, \mathbf{z}) = \prod_t p(z_t \mid z_{t-1}) p(x_t \mid z_t)$, and we aim to optimize the data likelihood $\log p(\mathbf{x})$. We begin by autoregressively decomposing the data likelihood as: $\log p(\mathbf{x}) = \sum_t \log p(x_t \mid x_{<t})$. For a given $t$, we evaluate the conditional likelihood $p(x_t \mid x_{<t})$ by inferring over two latent states only: $z_{t-1}$ and $z_t$, as they will naturally make belief states appear for times $t-1$ and $t$:

$$\log p(x_t \mid x_{<t}) \geq \mathop{\mathbb{E}}_{(z_{t-1}, z_t) \sim q(z_{t-1}, z_t \mid x_{\leq t})} \Big[ \log p(x_t \mid z_{t-1}, z_t, x_{<t}) + \log p(z_{t-1}, z_t \mid x_{<t})$$
$$- \log q(z_{t-1}, z_t \mid x_{\leq t}) \Big]. \tag{3}$$

Because of the Markov assumptions underlying the state-space model, we can simplify $p(x_t \mid z_{t-1}, z_t, x_{<t}) = p(x_t \mid z_t)$ and decompose $p(z_{t-1}, z_t \mid x_{<t}) = p(z_{t-1} \mid x_{<t}) p(z_t \mid z_{t-1})$. Next, we choose to decompose $q(z_{t-1}, z_t \mid x_{\leq t})$ as a belief over $z_t$ and a one-step smoothing distribution over $z_{t-1}$: $q(z_{t-1}, z_t \mid x_{\leq t}) = q(z_t \mid x_{\leq t}) q(z_{t-1} \mid z_t, x_{\leq t})$. We obtain the following belief-based

ELBO for state-space models:

$$\log p(x_t \mid x_{<t}) \geq \mathop{\mathbb{E}}_{(z_{t-1}, z_t) \sim q(z_{t-1}, z_t \mid x_{\leq t})} \Big[ \log p(x_t \mid z_t) + \log p(z_{t-1} \mid x_{<t}) + \log p(z_t \mid z_{t-1})$$

$$- \log q(z_t \mid x_{\leq t}) - \log q(z_{t-1} \mid z_t, x_{\leq t}) \Big]. \tag{4}$$

Both quantities $p(z_{t-1} \mid x_{\leq t-1})$ and $q(z_t \mid x_{\leq t})$ represent the belief state of the model at different times, so at this stage we approximate them with the same distribution $p_B(z \mid b)$, with $b_t = f(b_{t-1}, x_t)$ representing the belief state code for $z_t$. Similarly, we represent the smoothing posterior over $z_{t-1}$ as $q(z_{t-1} \mid z_t, b_{t-1}, b_t)$. We obtain the following loss:

$$-\mathcal{L} = \mathop{\mathbb{E}}_{\substack{z_t \sim p_B(z_t \mid b_t) \\ z_{t-1} \sim q(z_{t-1} \mid z_t, b_t, b_{t-1})}} \Big[ \log p(x_t \mid z_t) + \log p_B(z_{t-1} \mid b_{t-1}) + \log p(z_t \mid z_{t-1})$$

$$- \log p_B(z_t \mid b_t) - \log q(z_{t-1} \mid z_t, b_{t-1}, b_t) \Big]. \tag{5}$$

We provide an intuition on the different terms of the ELBO in the next section.

## 4   TD-VAE AND JUMPY STATE MODELING

The model derived in the previous section expresses a state model $p(z_t \mid z_{t-1})$ that describes how the state of the world evolves from one time step to the next. However, in many applications, the relevant timescale for planning may not be the one at which we receive observations and execute simple actions. Imagine for example planning for a trip abroad; the different steps involved (discussing travel options, choosing a destination, buying a ticket, packing a suitcase, going to the airport, and so on), all occur at vastly different time scales (potentially months in the future at the beginning of the trip, and days during the trip). Certainly, making a plan for this situation does not involve making second-by-second decisions. This suggests that we should look for models that can imagine future states directly, without going through all intermediate states.

Beyond planning, there are several other reasons that motivate modeling the future directly. First, training signal coming from the future can be stronger than small changes happening between time steps. Second, the behavior of the model should ideally be independent from the underlying temporal sub-sampling of the data, if the latter is an arbitrary choice. Third, jumpy predictions can be computationally efficient; when predicting several steps into the future, there may be some intervals where the prediction is either easy (e.g. a ball moving straight), or the prediction is complex but does not affect later time steps — which Neitz et al. (2018) call inconsequential chaos.

There is a number of research directions that consider temporal jumps. Koutnik et al. (2014) and Chung et al. (2016) consider recurrent neural network with skip connections, making it easier to bridge distant timesteps. Buesing et al. (2018) temporally sub-sample the data and build a jumpy model (for fixed jump size) of this data; but by doing so they also drop the information contained in the skipped observations. Neitz et al. (2018) and Jayaraman et al. (2018) predict sequences with variable time-skips, by choosing as target the most predictable future frames. They predict the observations directly without learning appropriate states, and only focus on nearly fully observed problems (and therefore do not need to learn a notion of belief state). For more general problems, this is a fundamental limitation, as even if one could in principle learn a jumpy observation model $p(x_{t+\delta} \mid x_{\leq t})$, it cannot be used recursively (feeding $x_{t+\delta}$ back to the RNN and predicting $x_{t+\delta+\delta'}$). This is because $x_{t+\delta}$ does not capture the full state of the system and so we would be missing information from $t$ to $t + \delta$ to fully characterize what happens after time $t + \delta$. In addition, $x_{t+\delta}$ might not be appropriate even as target, because some important information can only be extracted from a number of frames (potentially arbitrarily separated), such as a behavior of an agent.

### 4.1   THE TD-VAE MODEL

Motivated by the model derived in section 3, we extend sequential TD-VAE to exhibit time abstraction. We start from the same assumptions and architectural form: there exists a sequence of states $z_1, \ldots, z_T$ from which we can predict the observations $x_1, \ldots, x_T$. A forward RNN encodes a belief state $b_t$

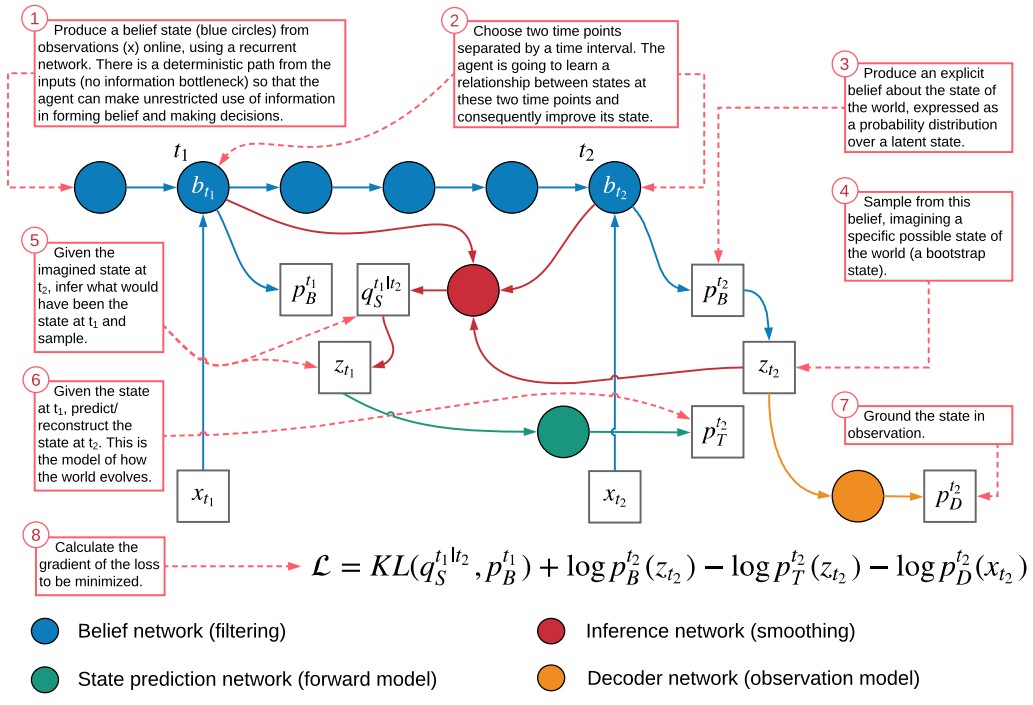

Figure 1: **Diagram of TD-VAE**. Follow the red panels for an explanation of the architecture. For succinctness, we use the notation $p_D$ to denote the decoder $p(x|z)$, $p_T$ to denote the transition distribution $p(s_{t_2}|s_{t_1})$, $q_S$ for the smoothing distribution and $p_B$ for the belief distribution.

from past observations $x_{\leq t}$. The main difference is that, instead of relating information known at times $t$ and $t+1$ through the states $z_t$ and $z_{t+1}$, we relate two distant time steps $t_1$ and $t_2$ through their respective states $z_{t_1}$ and $z_{t_2}$, and we learn a jumpy, state-to-state model $p(z_{t_2} \mid z_{t_1})$ between $z_{t_1}$ and $z_{t_2}$. Following equation 5, the negative loss for the TD-VAE model is:

$$\mathcal{L}_{t_1, t_2} = \mathop{\mathbb{E}}_{(z_{t_1}, z_{t_2}) \sim q(z_{t_1}, z_{t_2} \mid b_{t_1}, b_{t_2})} \left[ \log p(x_{t_2} \mid z_{t_2}) + \log p_B(z_{t_1} \mid b_{t_1}) + \log p(z_{t_2} \mid z_{t_1}) \right.$$

$$\left. - \log p_B(z_{t_2} \mid b_{t_2}) - \log q(z_{t_1} \mid z_{t_2}, b_{t_1}, b_{t_2}) \right] \tag{6}$$

To train this model, one should choose the distribution of times $t_1, t_2$; for instance, $t_1$ can be chosen uniformly from the sequence, and $t_2 - t_1$ uniformly over some finite range $[1, D]$; other approaches could be investigated. Figure 1 describes in detail the computation flow of the model.

Finally, it would be desirable to model the world with different hierarchies of state, the higher-level states predicting the same-level or lower-level states, and ideally representing more invariant or abstract information. For this reason, we also develop stacked (hierarchical) version of TD-VAE, which uses several layers of latent states. Hierarchical TD-VAE is detailed in the appendix.

## 4.2 INTUITION BEHIND TD-VAE

In this section, we provide a more intuitive explanation behind the computation and loss of the model. Assume we want to predict a future time step $t_2$ from all the information we have up until time $t_1$. All relevant information up until time $t_1$ (respectively $t_2$) has been compressed into a code $b_{t_1}$ (respectively $b_{t_2}$). We make an observation $x_t$ of the world[2] at every time step $t$, but posit the existence of a state $z_t$ which fully captures the full condition of the world at time $t$.

Consider an agent at the current time $t_2$. At that time, the agent can make a guess of what the state of the world is by sampling from its belief model $p_B(z_{t_2} \mid b_{t_2})$. Because the state $z_{t_2}$ should entail

---

[2]In RL, this observation may include the reward and previous action.

the corresponding observation $x_{t_2}$, the agent aims to maximize $p(x_{t_2} \mid z_{t_2})$ (first term of the loss), with a variational bottleneck penalty $-\log p(z_{t_2} \mid b_{t_2})$ (second term of the loss) to prevent too much information from the current observation $x_{t_2}$ from being encoded into $z_{t_2}$. Then follows the question 'could the state of the world at time $t_2$ have been predicted from the state of the world at time $t_1$?'. In order to ascertain this, the agent must estimate the state of the world at time $t_1$. By time $t_2$, the agent has aggregated observations between $t_1$ and $t_2$ that are informative about the state of the world at time $t_1$, which, together with the current guess of the state of the world $z_{t_2}$, can be used to form an ex post guess of the state of the world. This is done by computing a smoothing distribution $q(z_{t_1} \mid z_{t_2}, b_{t_1}, b_{t_2})$ and drawing a corresponding sample $z_{t_1}$. Having guessed states of the world $z_{t_1}$ and $z_{t_2}$, the agent optimizes its predictive jumpy model of the world state $p(z_{t_2} \mid z_{t_1})$ (third term of the loss). Finally, it should attempt to see how predictable the revealed information was, or in other words, to assess whether the smoothing distribution $q(z_{t_1} \mid z_{t_2}, b_{t_2})$ could have been predicted from information only available at time $t_1$ (this is indirectly predicting $z_{t_2}$ from the state of knowledge $b_{t_1}$ at time $t_1$ - the problem we started with). The agent can do so by minimizing the KL between the smoothing distribution and the belief distribution at time $t_1$: $\mathbf{KL}(q(z_{t_1} \mid z_{t_2}, b_{t_1}, b_{t_2}) \,\|\, p(z_{t_1} \mid b_{t_1}))$ (fourth term of the loss). Summing all the losses described so far, we obtain the TD-VAE loss.

### 4.3 CONNECTION WITH TEMPORAL-DIFFERENCE LEARNING

In reinforcement learning, the state of an agent represents a belief about the sum of discounted rewards $R_t = \sum_\tau r_{t+\tau}\gamma^\tau$. In the classic setting, the agent only models the mean of this distribution represented by the value function $V_t$ or action dependent Q-function $Q_t^a$ (Sutton & Barto, 1998). Recently in (Bellemare et al., 2017), a full distribution over $R_t$ has been considered. To estimate $V_{t_1}$ or $Q_{t_1}^a$ at time $t_1$, one does not usually wait to get all the rewards to compute $R_{t_1}$. Instead, one uses an estimate at some future time $t_2$ as a bootstrap to estimate $V_{t_1}$ or $Q_{t_1}^a$ (temporal difference).

In our case, the model expresses a *belief* $p_B(z_t \mid b_t)$ about possible future *states* instead of the *sum* of discounted *rewards*. The model trains the belief $p_B(z_{t_1} \mid b_{t_1})$ at time $t_1$ using belief $p_B(z_{t_2} \mid b_{t_2})$ at some time $t_2$ in the future. It accomplishes this by (variationally) auto-encoding a sample $z_{t_2}$ of the future state into a sample $z_{t_1}$, using the approximate posterior distribution $q(z_{t_1} \mid z_{t_2}, b_{t_1}, b_{t_2})$ and the decoding distribution $p(z_{t_2} \mid z_{t_1})$. This auto-encoding mapping translates between states at $t_1$ and $t_2$, forcing beliefs at the two time steps to be consistent. Sample $z_{t_1}$ forms the target for training the belief $p_B(z_{t_1} \mid b_{t_1})$, which appears as a prior distribution over $z_{t_1}$.

## 5 EXPERIMENTS.

The first experiment using sequential TD-VAE, which enables a direct comparison to related algorithms for training state-space models. Subsequent experiments use the full TD-VAE model.

### 5.1 PARTIALLY OBSERVED MINIPACMAN

We use a partially observed version of the MiniPacman environment (Racanière et al., 2017), shown in Figure 2. The agent (Pacman) navigates a maze, and tries to eat all the food while avoiding being eaten by a ghost. Pacman sees only a $5 \times 5$ window around itself. To achieve a high score, the agent needs to form a belief state that captures memory of past experience (e.g. which parts of the maze have been visited) and uncertainty on the environment (e.g. where the ghost might be).

We evaluate the performance of sequential (non-jumpy) TD-VAE on the task of modeling a sequence of the agent's observations. We compare it with two state-space models trained using the standard ELBO of equation 1:

- A **filtering model** with encoder $q(\mathbf{z} \mid \mathbf{x}) = \prod_t q(z_t \mid z_{t-1}, b_t)$, where $b_t = \mathrm{RNN}(b_{t-1}, x_t)$.
- A **mean-field model** with encoder $q(\mathbf{z} \mid \mathbf{x}) = \prod_t q(z_t \mid b_t)$, where $b_t = \mathrm{RNN}(b_{t-1}, x_t)$.

Figure 2 shows the ELBO and estimated negative log probability on a test set of MiniPacman sequences for each model. TD-VAE outperforms both baselines, whereas the mean-field model is the least well-performing. We note that $b_t$ is a belief state for the mean-field model, but not for the filtering model; the encoder of the latter explicitly depends on the previous latent state $z_{t-1}$, hence $b_t$

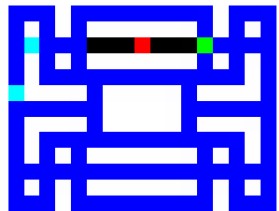

|                  | ELBO                  | $-\log p(\mathbf{x})$ (est.) |
|------------------|-----------------------|------------------------------|
| Filtering model  | $0.1169 \pm 0.0003$   | $0.0962 \pm 0.0007$          |
| Mean-field model | $0.1987 \pm 0.0004$   | $0.1678 \pm 0.0010$          |
| TD-VAE           | $\mathbf{0.0773 \pm 0.0002}$ | $\mathbf{0.0553 \pm 0.0006}$ |

Figure 2: **MiniPacman**. **Left**: A full frame from the game (size $15 \times 19$). Pacman (green) is navigating the maze trying to eat all the food (blue) while being chased by a ghost (red). **Top right**: A sequence of observations, consisting of consecutive $5 \times 5$ windows around Pacman. **Bottom right**: ELBO and estimated negative log probability on a test set of MiniPacman sequences. Lower is better. Log probability is estimated using importance sampling with the encoder as proposal.

Figure 3: **Moving MNIST**. **Left**: Rows are example input sequences. **Right**: Jumpy rollouts from the model. We see that the model is able to roll forward by skipping frames, keeping the correct digit and the direction of motion.

is not its sufficient statistics. This comparison shows that naively restricting the encoder in order to obtain a belief state hurts the performance significantly; TD-VAE overcomes this difficulty.

## 5.2 MOVING MNIST

In this experiment, we show that the model is able to learn the state and roll forward in jumps. We consider sequences of length 20 of images of MNIST digits. For each sequence, a random digit from the dataset is chosen, as well as the direction of movement (left or right). At each time step, the digit moves by one pixel in the chosen direction, as shown in Figure 3. We train the model with $t_1$ and $t_2$ separated by a random amount $t_2 - t_1$ from the interval $[1, 4]$. We would like to see whether the model at a given time can roll out a simulated experience in time steps $t_1 = t + \delta_1$, $t_2 = t_1 + \delta_2, \ldots$ with $\delta_1, \delta_2, \ldots > 1$, without considering the inputs in between these time points. Note that it is not sufficient to predict the future inputs $x_{t_1}, \ldots$ as they do not contain information about whether the digit moves left or right. We need to sample a state that contains this information.

We roll out a sequence from the model as follows: (a) $b_t$ is computed by the aggregation recurrent network from observations up to time $t$; (b) a state $z_t$ is sampled from $p_B(z_t \mid b_t)$; (c) a sequence of states is rolled out by repeatedly sampling $z \leftarrow z' \sim p(z' \mid z)$ starting with $z = z_t$; (d) each $z$ is decoded by $p(x \mid z)$, producing a sequence of frames. The resulting sequences are shown in Figure 3. We see that indeed the model can roll forward the samples in steps of more than one elementary time step (the sampled digits move by more than one pixel) and that it preserves the direction of motion, demonstrating that it rolls forward a state.

## 5.3 NOISY HARMONIC OSCILLATOR

We would like to demonstrate that the model can build a state even when little information is present in each observation, and that it can sample states far into the future. For this we consider a 1D sequence obtained from a noisy harmonic oscillator, as shown in Figure 4 (first and fourth rows). The frequencies, initial positions and initial velocities are chosen at random from some range. At every update, noise is added to the position and the velocity of the oscillator, but the energy is approximately preserved. The model observes a noisy version of the current position. Attempting to predict the input, which consists of one value, 100 time steps in the future would be uninformative; such a

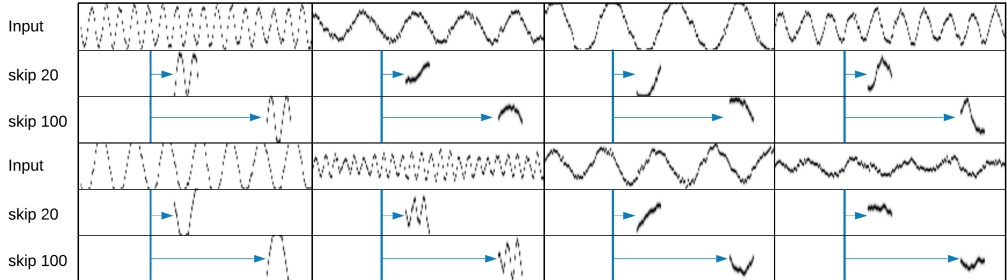

Figure 4: **Skip-state prediction for 1D signal**. The input is generated by a noisy harmonic oscillator. Rollouts consist of (a) a jumpy state transition with either $dt = 20$ or $dt = 100$, followed by 20 state transitions with $dt = 1$. The model is able to create a state and predict it into the future, correctly predicting frequency and magnitude of the signal.

prediction wouldn't reveal what the frequency or the magnitude of the signal is, and because the oscillator updates are noisy, the phase information would be nearly lost. Instead, we should try to predict as much as possible about the state, which consists of frequency, magnitude and position, and it is only the position that cannot be accurately predicted.

The aggregation RNN is an LSTM; we use a hierarchical TD-VAE with two layers, where the latent variables in the higher layer are sampled first, and their results are passed to the lower layer. The belief, smoothing and state-transition distributions are feed-forward networks, and the decoder simply extracts the first component from the $z$ of the first layer. We also feed the time interval $t_2 - t_1$ into the smoothing and state-transition distributions. We train on sequences of length 200, with $t_2 - t_1$ taking values chosen at random from $[1, 10]$ with probability 0.8 and from $[1, 120]$ with probability 0.2.

We analyze what the model has learned as follows. We pick time $t_1 = 60$ and sample $z_{t_1} \sim p_B(z_{t_1} \mid b_{t_1})$. Then, we choose a time interval $\delta_t \in \{20, 100\}$ to skip, sample from the forward model $p(z_2 \mid z_1, \delta_t)$ to obtain $z_{t_2}$ at $t_2 = t_1 + \delta_t$. To see the content of this state, we roll forward 20 times with time step $\delta = 1$ and plot the result, shown in Figure 4. We see that indeed the state $z_{t_2}$ is predicted correctly, containing the correct frequency and magnitude of the signal. We also see that the position (phase) is predicted well for $dt = 20$ and less accurately for $dt = 100$ (at which point the noisiness of the system makes it unpredictable).

Finally, we show that TD-VAE training can improve the quality of the belief state. For this experiment, the harmonic oscillator has a different frequency in each interval $[0, 10), [10, 20), [20, 120), [120, 140)$. The first three frequencies $f_1, f_2, f_3$ are chosen at random. The final frequency $f_4$ is chosen to be one fixed value $f_a$ if $f_1 > f_2$ and another fixed value $f_b$ otherwise ($f_a$ and $f_b$ are constants). In order to correctly model the signal in the final time interval, the model needs to learn the relation between $f_1$ and $f_2$, store it over length of 100 steps, and apply it over a number of time steps (due to

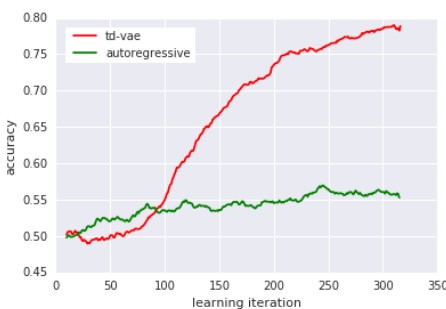

the noise) in the final interval. To test whether the belief state contains the information about this relationship, we train a binary classifier from the belief state to the final frequency $f_4$ at points just before the final interval. We compare two models with the same recurrent architecture (an LSTM), but trained with different objective: next-step prediction vs TD-VAE loss. The figure on the right shows the classification accuracy for the two methods, averaged over 20 runs. We found that the longer the separating time interval (containing frequency $f_3$) and the smaller the size of the LSTM, the better TD-VAE is compared to next-step predictor.

## 5.4 DeepMind Lab environment

In the final experiment, we analyze the model on a more visually complex domain. We use sequences of frames seen by an agent solving tasks in the DeepMind Lab environment (Beattie et al., 2016). We aim to demonstrate that the model holds explicit beliefs about various possible futures, and that

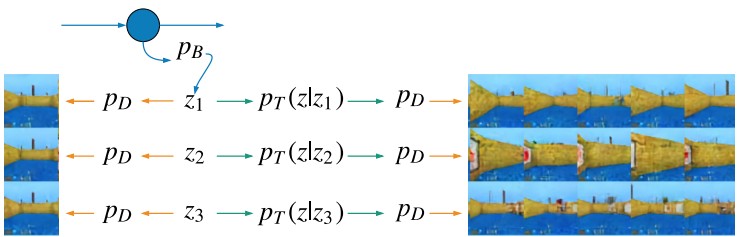

Figure 5: **Beliefs of the model**. **Left**: Independent samples $z_1, z_2, z_3$ from current belief; all 3 decode to roughly the same frame. **Right**: Multiple predicted futures for each sample. The frames are similar for each $z_i$, but different across $z_i$'s.

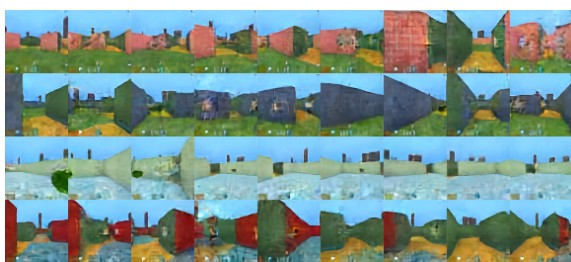

Figure 6: **Rollout from the model**. The model was trained on steps uniformly distributed in $[1, 5]$. The model is able to create forward motion that skips several time steps.

it can roll out in jumps. We suggest functional forms inspired by convolutional DRAW: we use convolutional LSTMs for all the circles in Figure 8 and make the model 16 layers deep (except for the forward updating LSTMs which are fully connected with depth 4).

We use time skips $t_2 - t_1$ sampled uniformly from $[1, 40]$ and analyze the content of the belief state $b$. We take three samples $z_1, z_2, z_3$ from $p_B(z \mid b)$, which should represent three instances of possible futures. Figure 5 (left) shows that they decode to roughly the same frame. To see what they represent about the future, we draw 5 samples $z_i^k \sim p(\hat{z} \mid z)$, $k = 1, \ldots, 5$ and decode them, as shown in Figure 5 (right). We see that for a given $i$, the predicted samples decode to similar frames (images in the same row). However $z$'s for different $i$'s decode to different frames. This means $b$ represented a belief about several different possible futures, while different $z_i$ each represent a single possible future.

Finally, we show what rollouts look like. We train on time separations $t_2 - t_1$ chosen uniformly from $[1, 5]$ on a task where the agent tends to move forward and rotate. Figure 6 shows 4 rollouts from the model. We see that the motion appears to go forward and into corridors and that it skips several time steps (real single step motion is slower).

## 6 CONCLUSIONS

In this paper, we argued that an agent needs a model that is different from an accurate step-by-step environment simulator. We discussed the requirements for such a model, and presented TD-VAE, a sequence model that satisfies all requirements. TD-VAE builds states from observations by bridging time points separated by random intervals. This allows the states to relate to each other directly over longer time stretches and explicitly encode the future. Further, it allows rolling out in state-space and in time steps larger than, and potentially independent of, the underlying temporal environment/data step size. In the future, we aim to apply TD-VAE to more complex settings, and investigate a number of possible uses in reinforcement learning such are representation learning and planning.

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

## A  TD-VAE AS A MODEL OF JUMPY OBSERVATIONS

In section 3, we derive an approximate ELBO which forms the basis of the training loss of the one-step TD-VAE. One may wonder whether a similar idea may underpin the training loss of the jumpy TD-VAE. Here we show how to modify the derivation to provide an approximate ELBO for a slightly different training regime.

Assume a sequence $(x_1, \ldots, x_T)$, and an arbitrary distribution $S$ over subsequences $\mathbf{x}_s = (x_{t_1}, \ldots, x_{t_n})$ of $\mathbf{x}$. For each time index $t_i$, we suppose a state $z_{t_i}$, and model the subsequence $\mathbf{x}_s$ with a jumpy state-space model $p(\mathbf{x}_s) = \prod_i p(z_{t_i}|z_{t_{i-1}})p(x_{t_i}|z_{t_i})$; denote $\mathbf{z}_s = (z_{t_1}, \ldots, z_{t_n})$ the state subsequence. We use the exact same machinery as the next-step ELBO, except that we enrich the posterior distribution over $\mathbf{z}_s$ by making it depend not only on observation subsequence $\mathbf{x}_s$, but on the entire sequence $\mathbf{x}$. This is possible because posterior distributions can have arbitrary contexts; the observations which are part of $\mathbf{x}$ but not $\mathbf{x}_s$ effectively serve as auxiliary variable for a stronger posterior. We use the full sequence $\mathbf{x}$ to form a sequence of belief states $b_t$ at all time steps. We use in particular the ones computed at the subsampled times $t_i$. By following the same derivation as the one-step TD-VAE, we obtain:

$$
\mathbb{E}_S\left[\log p(x_{t_1}, \ldots, x_{t_n})\right] \geq \mathbb{E}_S\left[\sum_i \mathop{\mathbb{E}}_{(z_{t_{i-1}}, z_{t_i}) \sim q}\left[\log p(x_{t_i} \mid z_{t_i}) + \log p(z_{t_{i-1}} \mid x_{<t})\right.\right.
$$
$$
+ \log p(z_{t_i} \mid z_{t_{i-1}}) - \log q(z_{t_i} \mid x_{\leq t})
$$
$$
\left.\left. - \log q(z_{t_{i-1}} \mid z_{t_i}, x_{\leq t})\right]\right]
$$

which, using the same belief approximations as the next step TD-VAE, becomes:

$$
-\mathcal{L} = \mathbb{E}_S\left[\sum_i \mathop{\mathbb{E}}_{\substack{z_{t_i} \sim p_B(z_{t_i}|b_{t_i}) \\ z_{t_{i-1}} \sim q(z_{t_{i-1}}|z_{t_i}, b_{t_i}, b_{t_{i-1}})}}\left[\log p(x_{t_i} \mid z_{t_i}) + \log p_B(z_{t_{i-1}} \mid b_{t_{i-1}}) + \log p(z_{t_i} \mid z_{t_{i-1}})\right.\right.
$$
$$
\left.\left. - \log p_B(z_{t_i} \mid b_{t_i}) - \log q(z_{t_{i-1}} \mid z_{t_i}, b_{t_{i-1}}, b_{t_i})\right]\right]
$$

which is the same loss as the TD-VAE for a particular choice of the sampling scheme $S$ (only sampling pairs).

## B  DERIVATION OF THE TD-VAE MODEL FROM ITS DESIRED PROPERTIES

In this section we start with a general recurrent variational auto-encoder and consider how the desired properties detailed in sections 1 and 2 constrain the architecture. We will find that these constraints in fact naturally lead to the TD-VAE model.

Let us first consider a relatively general form of temporal variational auto-encoder. We consider recurrent models where the same module is applied at every step, and where outputs are sampled one at a time (so that arbitrarily long sequences can be generated). A very general form of such an architecture consist of forward-backward encoder RNNs and a forward decoder RNN (Figure 7) but otherwise allowing for all the connections. Several works (Chung et al., 2015; Lee et al., 2018; Archer et al., 2015; Fraccaro et al., 2016; Liu et al., 2017; Goyal et al., 2017; Buesing et al., 2018; Serban et al., 2017) fall into this framework.

Now let us consider our desired properties.

In order to sample forward in latent space, the encoder must not feed into the decoder or the prior of the latent variables, since observations are required to compute the encoded state, and we would therefore require the sampled observations to compute the distribution over future states and observations.

We next consider the constraint of computing a belief state $b_t$. The belief state $b_t$ represents the state of knowledge up to time $t$, and therefore cannot receive an input from the backwards decoder.

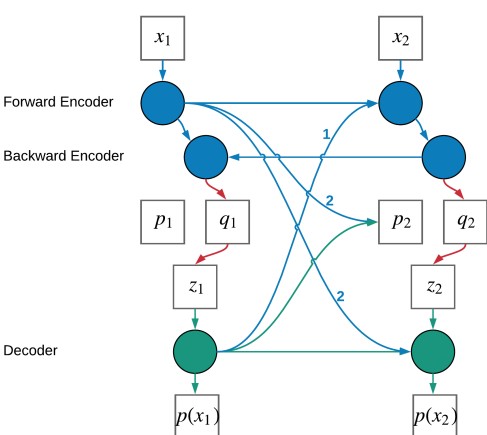

Figure 7: **Recurrent variational auto-encoder**. General recurrent variational auto-encoder, obtained by imposing recurrent structure, forward sampling and allowing all potential connections. Note that the encoder can have several alternating layers of forward and backward RNNs. Also note that the connection 1 has to be absent if the backwards encoder is used. Possible skip connections are not shown as they can directly be implemented in the RNN weights. If connections 2 are absent, the model is capable of forward sampling in latent space without going back to observations.

Furthermore, $b_t$ should have an unrestricted access to information; it should ideally not be disturbed by sampling (two identical agents with the same information should compute the same information; this will not be the case if the computation involves sampling), nor go through information bottlenecks. This suggests using the forward encoder for computing the belief state.

Given the use of a decoder RNN, the information needed to predict the future could be stored in the decoder state, which may prevent the encoder from storing the full state information (in other words, the information contained in $x_1, \ldots, x_{t+1}$ about the state $z_{t+1}$ could be partially stored in the decoder state and previous sample $z_t$). This presents two options: the first is to make the prior $p(z_{t+1}|.)$ and the reconstruction $p(x_t|.)$ depend only on $z_t$, i.e. to only consider distributions $p(z_{t+1}|z_t)$ and $p(x_t|z_t)$. The second is to include the decoder state in the belief state (together with the encoder state). We will choose the former option, as we our next constraint will invalidate the latter option.

Next, we argue that smoothing, or the dependence of posterior on the future, is an important property that should be part of our model. As an example, imagine a box that can contain two items $A$ and $B$ and two time points: $t_1$ before opening the box, when we don't know the content of the box, and $t_2$ after opening it. We would want our latent variable to represent the content of the box. The perfect model of the content of the box is that the content doesn't change (the same object is in the box before and after opening it). Now imagine $B$ is in the box. Our belief at $t_2$ is high for $B$ but our belief at $t_1$ is uncertain. If we sample this belief at $t_1$ without considering $t_2$ we would sample $A$ half of the time. However, then we would be learning a wrong model of the world: that $A$ goes to $B$. To solve this problem, we should sample $t_2$ first and then, given this value, sample $t_1$.

Smoothing requires the use of the backward encoder; this prevents the use of the decoder state as part of our belief state, since the decoder has access to the encoder, and the encoder depends on the future. We therefore require a latent-to-latent model $p(z_{t+1}|z_t)$.

We are therefore left with a forward encoder which ideally computes the belief state, a backwards encoder which - with the forward encoder - compute posteriors over states, and a state-to-state forward model. The training of the backwards encoder will be induced by its use as a posterior in the state-space model. How do then make sure the forward encoder is in fact trained to contain the belief state? To do so, we will force $p_B(z_t | b_t)$ to be close to the posterior by using a KL term between prior belief and posterior belief.

Before detailing the KL term, we need to consider how to practically run the backwards decoder. Ideally, we would like to train the model in a nearly forward fashion, for arbitrary long sequences.

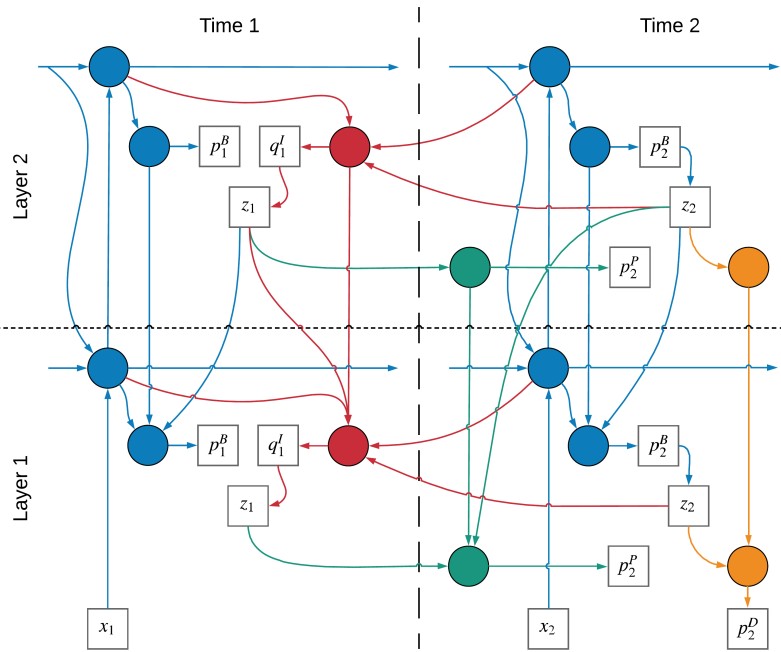

Figure 8: **Deep version of the model from Figure 1**. A deep version of the model is formed by creating a layer similar to the shallow model of Figure 1 and replicating it. Both sampling and inference proceed downwards through the layers. Circles have the same meaning as in Figure 1 and are implemented using neural networks, such as LSTMs.

This prevents running the backwards inference from the end of the sequence. However if we assume that $p_B$ represents our best belief about the future, we can take a sample from it as an instance of the future: $z_{t_2} \sim p_B(z_{t_2}|b_{t_2})$. It forms a type of bootstrap information. Then we can go backwards and infer what would the world have looked like given this future (e.g. the object $B$ was still in the box even if we don't see it). Using VAE training, we sample $z_1$ from its posterior $q(z_{t_1}|z_{t_2}, b_{t_2}, b_{t_1})$ (the conditioning variables are the ones we have available locally), using $p_B(z_{t_1}|b_{t_1})$ as prior. Conversely, for $t_2$, we sample from $p_B(z_{t_2}|b_{t_2})$ as posterior, but with $p(z_{t_2}|z_{t_1})$ as prior. We therefore obtain the VAE losses $\log q(z_1|z_2, s_1, s_2) - \log p_B(z_1|s_1)$ at $t_1$ and $\log p_B(z_2|s_2) - \log p_P(z_2|z_1)$ at $t_2$. In addition we have the reconstruction term $p_D(x_2|z_2)$ that grounds the latent in the input. The whole algorithm is presented in the Figure 1.

## C   Hierarchical Model

In the main paper we detailed a framework for learning models by bridging two temporally separated time points. It would be desirable to model the world with different hierarchies of state, the higher-level states predicting the same-level or lower-level states, and ideally representing more invariant or abstract information. In this section we describe a stacked (hierarchical) version of the model.

The first part to extend to $L$ layers is the RNN that aggregates observations to produce the belief state $b$. Here we simply use a deep LSTM, but with layer $l$ receiving inputs also from layer $l + 1$ from the previous time step. This is so that the higher layers can influence the lower ones (and vice versa). For $l = 1, \ldots, L$:

$$b_t^l = \text{RNN}(b_t^l, b_t^{l-1}, b_{t-1}^{l+1}, x_t) \tag{7}$$

and setting $b_0 = b_L$ and $b_{L+1} = \emptyset$.

We create a deep version of the belief part of the model by stacking the shallow one, as shown in Figure 8. In the usual spirit of deep directed models, the model samples downwards, generating higher level representations before the lower level ones (closer to pixels). The model implements deep inference, that is, the posterior distribution of one layer depends on the samples from the posterior

distribution in previously sampled layers. The order of inference is a design choice, and we use the same direction as that of generation, from higher to lower layers, as done for example by Gregor et al. (2016); Kingma et al. (2016); Rasmus et al. (2015). We implement the dependence of various distributions on latent variables sampled so far using a recurrent neural network that summarizes all such variables (in a given group of distributions). We don't share the weights between different layers. Given these choices, we can allow all connections consistent with the model. Next we describe the functional forms used in our model.

## D  FUNCTIONAL FORMS AND PARAMETER CHOICES

Here we describe the functional forms used in more detail. We start with those used for the harmonic oscillator experiments. Let $x_t, t = 1, \ldots, T$ be the input sequence. The belief state network (both is a standard LSTM network: $b_t, c_t = \mathrm{LSTM}(x_t, b_{t-1}, c_{t-1})$. For any arbitrary context $x$, we denote $D$ the map from $x$ to a normal distribution with mean $\mu(x)$ and log-standard deviation $\log \sigma(x)$, where $[\mu, \log \sigma] = W_3 \tanh(W_1 x + B_1)\sigma(W_2 x + B_2) + B_3$, with $W_1, W_2, W_3$ as weight matrices and $B_1, B_2, B_3$ as biases. We use the letter $D$ for all such maps (even when they don't share weights); weights are shared if the contexts are identical except for the time index. Consider the update for a given pair of time points $t_1 < t_2$. We use a two-layer hierarchical TD-VAE. A variable $v$ at layer $l$ and time $t$ is denoted $v_t^l$. Beliefs are time $t_1$ and $t_2$ are denoted $b_{t_1}, b_{t_2}$. The set of equations describing the system are as follows.

$$
\begin{aligned}
z_{t_2}^2 &\sim p_B^{t_2^2} = D(b_{t_2}) \\
z_{t_2}^1 &\sim p_B^{t_2^1} = D(b_{t_2}, z_{t_2}^2) \\
z_{t_2} &= [z_{t_2}^1, z_{t_2}^2] \\
p_B^{t_1^2} &= D(b_{t_1}) \\
z_{t_1}^2 &\sim q_S^{t_1^2|t_2} = D(b_{t_1}, z_{t_2}, \delta_t) \\
p_B^{t_1^1} &= D(b_{t_1}, z_{t_1}^2) \\
z_{t_1}^1 &\sim q_S^{t_1^1|t_2} = D(b_{t_1}, z_{t_2}, z_{t_1}^2, \delta_t) \\
z_{t_1} &= [z_{t_1}^1, z_{t_1}^2] \\
p_T^{t_2^2|t_1} &= D(z_{t_1}, \delta_t) \\
p_T^{t_2^1|t_1} &= D(z_{t_1}, z_{t_2}^2, \delta_t) \\
p_D &= N(z_{t_2}, \sigma) \\
L_{t_1}^2 &= KL(q_S^{t_1^2|t_2} | p_B^{t_1^2}) \\
L_{t_1}^1 &= KL(q_S^{t_1^1|t_2} | p_B^{t_1^1}) \\
L_{t_2}^2 &= \log p_B^{t_2^2}(z_{t_2}^2) - \log p_T^{t_2^2|t_1}(z_{t_2}^2) \\
L_{t_2}^1 &= \log p_B^{t_2^1}(z_{t_2}^1) - \log p_T^{t_2^1|t_1}(z_{t_2}^1) \\
L_x &= -\log(p_D(x_{t_2})) \\
L &= L_2^1 + L_1^1 + L_2^2 + L_1^2 + L_x
\end{aligned}
\tag{8}
$$

The hidden layer of the $D$ maps is 50; the size of each $z_t^l$ is 8. Belief states have size 50. We use the Adam optimizer with learning rate 0.0005.

The same network works for the MNIST experiment with the following modifications. Observations are pre-processed by a two hidden layer MLP with ReLU nonlinearity. The decoder $p_D$ also have a two layer MLP, which outputs the logits of a Bernoulli distribution. $\delta_t$ was not passed as input to any network.

For the DeepMind Lab experiments, all the circles in Figure 8 are LSTMs. Blue circles are fully connected LSTM, the others are all convolutional LSTM. We use a fully connected LSTM of size $512$ and convolutional layers of size $4 \times 4 \times 256$. All kernel sizes are $3 \times 3$. The decoder layer has an extra canvas layer, similar to DRAW.

