# OpenReview forum: "Temporal Difference Variational Auto-Encoder"
_ICLR.cc/2019/Conference_

### Official Review · AnonReviewer3 · 2018-11-01
**TD-VAE**

**Rating:** 7
**Confidence:** 5

**Review:**

There are several ingredients in this paper that I really liked. For example, (1) the notion that an agent should build a deterministic function of the past which implicitly captures the belief (the uncertainty or probability distribution about the state), by opposition for example to sampling trajectories to capture uncertainty, (2) modelling the world's dynamic in a learned encoded state-space (by opposition to the sensor space), (3) instead of modeling next-step probabilities p(z(t+1)|z(t)), model 'jumpy transitions' p(z(t+delta)|z(t)) to avoid unrolling at the finest time scale.

Now for the weak points:
(a) the justification for the training loss was not completely clear to me, although I can see that it has a variational flavor
(b) there is no discussion of the issue that we can't get a straightforward decomposition of the joint probability over the data sequence according to next-step probabilities via the chain rule of probabilities, so we don't have a clear way to compare the TD-VAE models with jumpy predictions against other more traditional models
(c) none of the experiments make comparisons against previously published models and quantitative results (admittedly because of (b) this may not be easy).

So I believe that the authors are onto a great direction of investigation, but the execution of the paper could be improved.

---

> ### Author Response · Authors · 2018-11-25
> **Re:**
>
> Thank you for your review and comments. We clarified our intuitive derivation of the loss in section A. It is indeed difficult to compare the jumpy TD-VAE model to other models, as there is little work that studies such models. We updated the appendix to explain how a model similar to jumpy TD-VAE provides an approximate ELBO to the ‘jumpy’ log likelihood log p(x_{t_1}, x_{t_2}, .. x_{t_n}). As for comparison to published models, we did compare the sequential TD-VAE elbo on the simple mini-pacman dataset to classical state-space models; we also compared the belief state obtained by training a TD-VAE on the oscillator network to a more classical lstm in a recurrent classification setup. Following the line of the thinking, we believe an appropriate way to compare similar models will be through the comparison of the different belief states they learn. We highlighted this in the text.

---

### Official Review · AnonReviewer2 · 2018-11-04
**Very strong**

**Rating:** 9
**Confidence:** 4

**Review:**

The authors propose TD-VAE to solve an important problem in agent learning, simulating the future by doing jumpy-rollouts in abstract states with uncertainty. The authors first formulate the sequential TD-VAE and then generalize it for jumpy rollouts. The proposed method is well evaluated for four tasks including high dimensional complex task.

Pros.
- Advancing a significant problem
- Principled and quite original modeling based on variational inference
- Rigorous experiments including complex high dimensional experiments
- Clear and intuitive explanation (but can be improved further)

Cons.
- Some details on the experiments are missing (due to page limit). It would be great to include these in the Appendix.
- It is a complex model. For reproducibility, detail specification on the hyperparameters and architecture will be helpful.

Minor comments
- Why q(z_{t-1}|z_t, b_{t-1}, b_t) depends both  b_{t-1}, b_t, not only b_t?
- The original model does not take the jump interval as input. Then, it is not clear how the jump interval is determined in p(z’|z)?

---

> ### Author Response · Authors · 2018-11-25
> **Re:**
>
> Thank you for the review and comments.
> Thanks for the suggestion - we added missing experiment details, network specifications and hyperparameters in the appendix.
> You are correct that q(z_{t-1}|z_t, b_{t-1}, b_t) does not need to depend on b_{t-1}, but it does not hurt to do so; we chose to do so in order to further facilitate the learning of b_{t-1}, but it may not have affected experiments.
> If the model does not take the jump interval as input, the model has to represent the jump size by way of a multimodal distribution over possible future events. One could imagine that one of the latent variables could be learned to correspond to dt.

---

### Official Review · AnonReviewer1 · 2018-11-06
**Nice and novel idea**

**Rating:** 8
**Confidence:** 4

**Review:**

This paper proposes the temporal difference variational auto-encoder framework, a sequential general model following the intuition of temporal difference learning in reinforcement learning. The idea is nice and novel, and I vote for acceptance.
1. The introduction of belief state in the sequential model is smart. How incorporate such technique in such an autoregressive model is not easy.
2. Fig 1 clearly explained the VAE process.
3. Four experiments demonstrated the main advantages of the proposed framework, including the effectiveness of proposed belief state construction and ability to jumpy rolling-out,


Other Comments and Questions:
1. Typo, p(s_{t_2}|s_{t_1}) in the caption of Fig 1.
2. Can this framework partially solve the exposure bias?
3. The author used uniform distribution for t_2 - t1, and from the ``NOISY HARMONIC OSCILLATOR`` we can indeed see larger interval will result in worse performance. However, the author also mentioned other distortion could be investigated, so I am wondering if the larger probability mass is put on larger dt, what the performance will become.
4. The code should be released. I think that it is a fundamental framework deserving further development  by other researchers.

---

> ### Author Response · Authors · 2018-11-25
> **Re:**
>
> Thank you for your thoughtful review and comments.
>
> Thanks for noticing the typo - we will fix it.
> Regarding the exposure bias - TD-VAE may indeed reduce exposure bias by generating faraway futures in fewer steps of generation. But we have not explicitly investigated that issue in the paper.
> Regarding the distribution of (t_2-t_1), for the noisy harmonic oscillator experiment we use a mixture of two uniform distributions, one with support [1,T], the second with support [1,T’], with T’>T. Since shorter time steps are easy to model, this served as a form of ‘curriculum’ for the jumpy model; this enables us to learn the state representation, which in turns facilitates learning the ‘jumpier’ transitions from [1,T’]. We clarify this in the text.  It is indeed likely that weighting [1,T'] more heavily would indeed improve the jumpier prediction.
> More general strategies could be adopted, for instance choosing jump sizes which make the jump easy to predict (as is suggested in Neitz et al. and Jayaraman et al.), or hard to predict (a form of prioritized replay for model learning), or any other criterion. We reserve the investigation of which scheme leads to the best model to future work.
> As for code, we will aim to release a simplified version of the code in the future.

---

### Public Comment · (anonymous) · 2019-01-23
**About DeepMind Lab experiments**

Thanks for your great work.

I have one question about DeepMind Lab experiments in this paper.
In Appendix D, you mentioned that p_D(x_{t_2}) is Bernoulli distribution and the log-likelihood is calculated using the logits outputted by the network in MNIST experiments.
Is it same in the DeepMind Lab experiments?
I think Normal distribution with a fixed variance is often used as decoder distribution in such color image generation, so if you used a different setting for DeepMind Lab experiments, I hope the setting is clearly written in the paper.

Thank you.

---

### Meta-Review · Area_Chair1 · 2018-12-14
**Original paper**

**Confidence:** 4
**Recommendation:** Accept (Oral)

**Metareview:**

The reviewers agree that this is a novel paper with a convincing evaluation.